# Modelling global trade with optimal transport

Thomas Gaskin ®[1,2,3] ✉, Guven Demirel[4], Marie-Therese Wolfram[5] & Andrew Duncan[3]

Global trade is shaped by a complex mix of factors beyond supply and demand, including tangible variables like transport costs and tariffs, as well as less quantifiable influences such as political and economic relations. Traditionally, economists model trade using gravity models, which rely on explicit covariates that might struggle to capture these subtler drivers of trade. In this work, we employ optimal transport and a deep neural network to learn a time-dependent cost function from data, without imposing a specific functional form. This approach consistently outperforms traditional gravity models in accuracy and has similar performance to three-way gravity models, while providing natural uncertainty quantification. Applying our framework to global food and agricultural trade, we show that low income countries experienced disproportionately higher increases in trade costs due to the war in Ukraine's impact on wheat markets. We also analyse the effects of free-trade agreements and trade disputes with China, as well as Brexit's impact on British trade with Europe, uncovering hidden patterns that trade volumes alone cannot reveal.

International trade serves as the backbone of the world economy, distributing goods and connecting markets through global logistics networks. Its dynamics are driven by numerous factors beyond mere supply and demand, such as tariffs, non-tariff policy barriers, political and economic tensions, and disruptions caused by accidents, conflicts, and civil wars. Among all traded commodities, agricultural and food products hold particular interest for policymakers and the general public due to their considerable volume, high trade value, and critical role in food security and resilience[1,2]. Consumer food prices are a product of all the complexly interwoven factors governing trade. However, they do not always directly reflect the ease of doing business between any two countries. For instance, in May 2020, China imposed an 80% tariff on Australian barley, leading to a major restructuring of global supply chains: Chinese demand was suddenly met from France, Canada, and Argentina, while Australia started exporting surplus barley e.g. to Saudi Arabia. Despite these

shifts, for the next five months the global barley price barely budged[3,4].

Modelling global trade has garnered much attention in the economics literature, with gravity models being the most widely used approach[5–9]. These models, named for their direct analogy to Newton's law of gravity, assume that the total trade $T_{ij}$ of a given commodity between two countries $i$ and $j$ is proportional to the total output $O_i$ of the source country and the total expenditure $E_j$ of the destination country, as well as being inversely related to a 'distance' between them:

$$T_{ij}(t) \sim \frac{O_i(t)E_j(t)}{C_{ij}(t)}. \tag{1}$$

This distance $C_{ij}$ comprises all factors that contribute to the ease of selling goods produced in one country to another, including transportation costs, information costs, and tariff and non-tariff barriers to

[1]Department of Methodology, London School of Economics and Political Science, London, UK. [2]Department of Applied Mathematics and Theoretical Physics, University of Cambridge, Cambridge, UK. [3]Department of Mathematics, Imperial College London, London, UK. [4]School of Business and Management, Queen Mary University of London, London, UK. [5]Mathematics Institute, University of Warwick, Coventry, UK. ✉e-mail: t.gaskin@lse.ac.uk

trade. Traditional gravity models use a set of covariates to estimate $C_{ij}$ as

$$\log C_{ij}(t) = \sum_k \alpha_k \pi_{i,k}(t) + \sum_l \beta_l \chi_{j,l}(t) + \sum_m \gamma_m \rho_{ij,m}(t), \quad (2)$$

where $\pi_i$ and $\chi_j$ are exogenous exporter and importer-side regressors[10], which can be specified as fixed effects, $\rho_{ij}$ are bilateral covariates, and $\alpha$, $\beta$, $\gamma$ are the coefficient vectors. Commonly used covariates include geographic proximity, the existence of trade agreements, colonial ties, tariffs, non-tariff barriers, or shared languages[11]. The structural gravity model corrects Eq. (1) with import and export multilateral resistance terms, which account for the relative nature of bilateral trade shares. This adjustment has been shown to align with various microeconomic models[7]. Gravity models have been widely used to study agrifood trade. For instance, ref. 12 estimates residual trade costs based on a micro-founded gravity equation, finding ad-valorem costs to be 60% higher in the Global South compared to the North. Studies have also investigated the impact of global and regional trade agreements[13,14] and the effect of eliminating tariffs[15,16].

The gravity-based approach is attractive to researchers due to its interpretability, mathematical simplicity, and consistency with various microeconomic theories[9]. However, it is not without its limitations. For one, multilateral trade resistance terms, central to the structural gravity model, are unobservable and must be estimated, often using fixed effects[11]. Elasticity and other key parameters are often unavailable at a granular level, requiring aggregation that can introduce bias[17]. The model's cost function also depends heavily on the choice of covariates and functional form, making specification crucial for interpreting results. In addition, unobservables—such as the subtle effects of changing political relations, public preferences, or aversions toward products from specific countries—are absorbed in the error term. Finally, while trade costs are generally asymmetric ($C_{ij} \neq C_{ji}$), commonly used covariates are not, making it difficult for a model to capture the inherent imbalances in trade relationships. See[9,11,18] for a deeper discussion of challenges and best practices.

In this work, we present a more general approach that dispenses with the use of covariates and a functional form, instead inferring the cost directly from data. Our method is based on the optimal transport (OT) framework[19], which generalises gravity-based models. In OT, trade flows are assumed to match supply and demand to minimise the overall cost. Mathematically, this is expressed as follows: let $\mathbf{C} \in \mathbb{R}^{m \times n}$ be a matrix quantifying the 'cost' (in a general sense) of moving goods from country $i$ to $j$. Given the supply vector $\boldsymbol{\mu} \in \mathbb{R}^m$ and the demand vector $\boldsymbol{\nu} \in \mathbb{R}^n$, the OT problem consists in finding a transport plan, i.e. a matrix $\mathbf{T} \in \mathbb{R}_+^{m \times n}$, with entries $T_{ij}$ modelling the total volume (or value) of transport from country $i$ to $j$, such that the total cost

$$c(\mathbf{T}) = \sum_{i,j} T_{ij} C_{ij} \quad (3)$$

is minimised. In addition, the marginal constraints

$$\sum_i T_{ij} = \boldsymbol{\mu}, \quad \sum_i T_{ij} = \boldsymbol{\nu} \quad (4)$$

must be satisfied, ensuring that demand and supply are met. It is advantageous to add a regularisation term to the cost, as it ensures existence of a unique solution and improves computational efficiency; the total cost then becomes

$$c_\varepsilon(\mathbf{T}) = c(\mathbf{T}) - \varepsilon \mathcal{H}(\mathbf{T}), \quad (5)$$

where $\mathcal{H}(\mathbf{T}) = -\sum_{ij} T_{ij}(\log T_{ij} - 1)$ denotes the negative entropy of $\mathbf{T}$ and $\varepsilon > 0$ is a regularisation parameter. It can be shown that the

solution will then be of the form

$$\mathbf{T} = \boldsymbol{\pi} e^{-\mathbf{C}/\varepsilon} \boldsymbol{\Omega}, \quad (6)$$

where $\boldsymbol{\Pi}$ and $\boldsymbol{\Omega}$ are diagonal scaling matrices which ensure that the marginal constraints hold (see 'Methods'). As described in ref. 20, gravity models can be reformulated as solutions of a regularised OT problem with an appropriate choice of parameters. While OT-based models might appear to suggest a centralised control of flows, its dual formulation admits an alternative, decentralised interpretation of importers seeking to minimise the cost of purchasing commodities and exporters seeking to maximise their profit (see 'Methods'). The solution at equilibrium coincides with the solution of the OT problem[21], which in its classic form (3)–(4) is well understood. This is less true for the corresponding inverse problem we are interested in, despite its mathematical and practical importance: given a (possibly noisy) observation of $\mathbf{T}$, $\boldsymbol{\mu}$, and $\boldsymbol{\nu}$, this problem consists in inferring the underlying cost $\mathbf{C}$. Maximum likelihood estimation of costs relates to the inverse OT problem; however, the specific parametrization of the cost matrix in gravity models demands careful estimator design. Moreover, zeros and heteroscedasticity in observed trade flows cause misspecification in gravity model estimation, affecting estimator performance (see ref. 22).

The inference methodology presented in this work is a deep learning approach to solve the inverse OT problem, based on recent work on neural parameter calibration[23,24]. We assume no underlying covariate structure of trade costs, but instead infer a general cost matrix $\mathbf{C}$, parametrised as a deep neural network, directly from data on trade flows. We train a neural network $u$ to recognise cost matrices from observations of transport plans for the global food and agricultural trade from 2000 to 2022 (the 'training data') by constraining it to satisfy Eq. (6). Put simply, this means fitting the mathematical OT equation to the data in such a way that the predicted cost matrices $\mathbf{C}(t)$ reproduce the observations $\mathbf{T}(t)$. The trained neural network then solves the inverse problem

$$\mathbf{C}(t) = u(\mathbf{T}(t)) \quad (7)$$

on the observations. Though its ability to generalise to new observations depends on the amount of training data, its performance on the training data itself does not. A probability density $\rho_C$ on the estimates is then naturally obtained as the pushforward measure

$$\rho_C = u_\# \rho_T, \quad (8)$$

where $\rho_T$ is the measure on $\mathbf{T}$. Additionally, we train a family of neural networks to capture the spread in cost matrices that optimally reproduce the transport plan (see 'Methods'). As we demonstrate, this approach produces trade flow estimates that are an order of magnitude more accurate than those of a traditional covariate-based gravity model.

The dataset under consideration was assembled by the Food and Agricultural Organisation of the United Nations (FAO), which provides global trade matrices for over 500 products on its portal[25]. Though extensive, many entries in the trade matrices are missing. Furthermore, the FAO reports two values for each bilateral flow $T_{ij}$: one reported by the exporter, and one reported by the importer. There is often a considerable discrepancy between the two, due to a multitude of epistemic factors the FAO lists in its accompanying report[26]. The uncertainty on our estimates naturally follows the uncertainty on the FAO data, without presupposing an underlying statistical model.

We apply our method to analyse global commodity flows from 2000 to 2022, examining the impacts of events, conflicts, trade agreements, and political changes on trade. The cost matrix uncovers economic effects that are not evident in trade volumes or retail prices

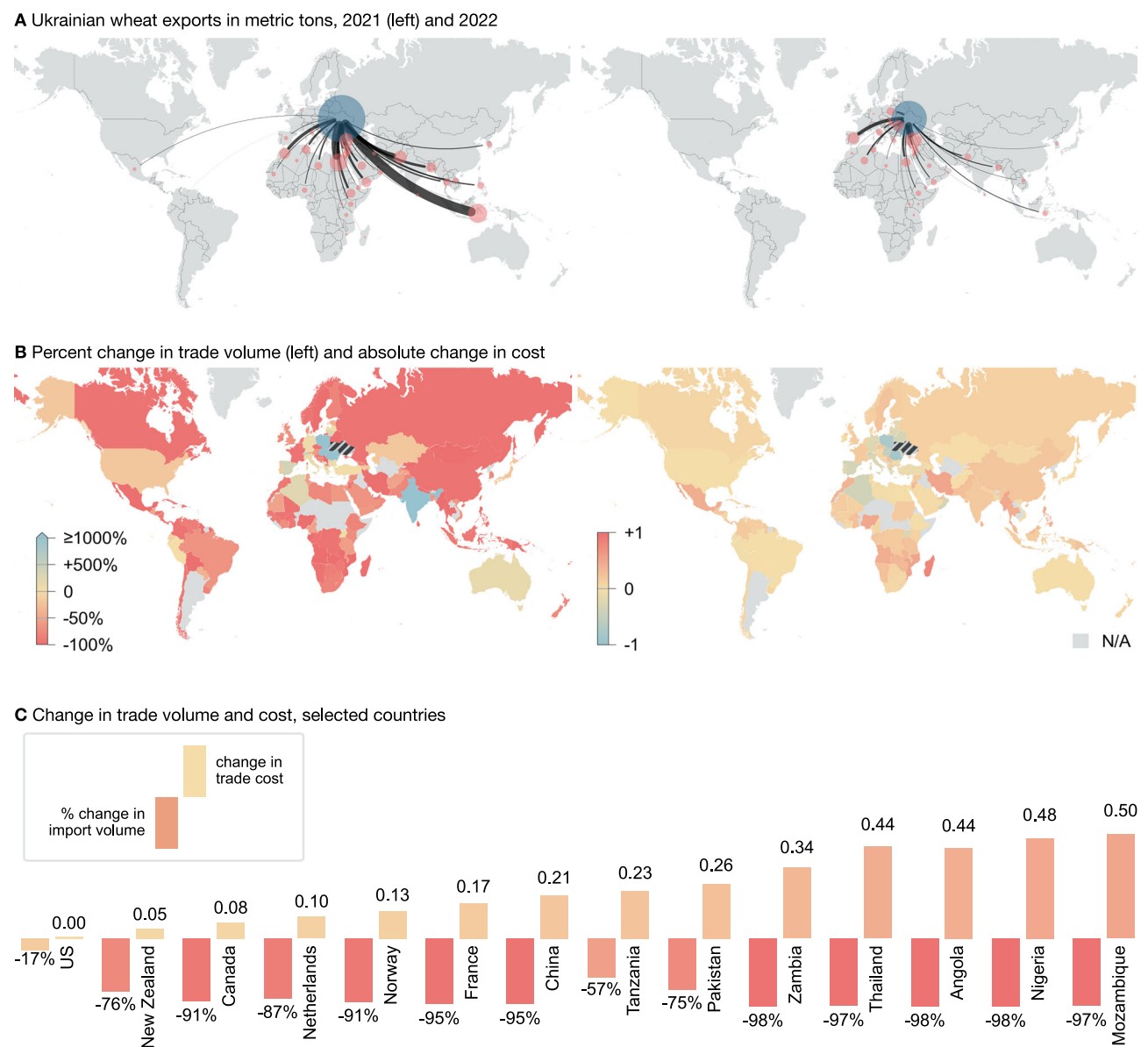

**Fig. 1 | Ukrainian wheat exports, 2021–2022. A** Network of Ukrainian exports, 2021 and 2022. Shown are the largest trading partners, making up 99% of Ukrainian exports. The blue node represents the total Ukrainian export volume (in metric tons), the red nodes are the import volumes. Edge widths represent the flow volume. **B** The change in trade volume (left) and trade cost (right) for the largest trading partners. The large relative increase in trade to India is small in absolute terms and statistically not significant. Countries in grey either have no trade data for the period shown, or are below the 99% pooling limit (see 'Methods'). **C** Percent change in trade volume (left bar) and change in trade cost (right bar) for selected countries.

alone. The article begins with a study of the war in Ukraine's impact on global wheat trade, followed by an analysis of free trade agreements and disputes in the Asia-Pacific, as well as the United Kingdom's exit from the European Union (Brexit). We demonstrate our method's ability to provide meaningful uncertainty quantification and compare it to a traditional gravity model, demonstrating superior prediction accuracy.

## Results

### Case study I: The impact of the war in Ukraine on wheat trade

The Russian Federation's invasion of Ukraine in early 2022 sent shock waves through global food markets[27]. Russia and Ukraine are two of the largest exporters of wheat, together accounting for almost 28% of global wheat exports in 2020. The blockade of trading routes through the Black Sea and the closure or destruction of ports in Mykolaiv and Kherson meant a drop in trade to the overwhelming majority of

Ukraine's export destinations, in some cases by as much as 100% (Fig. 1A, B). An increase of wheat exports only occurred to Europe, most notably to Poland, Spain, Slovakia, Romania, as well as to Algeria and Türkiye. However, our analysis indicates that, although trade shrank across the globe, the accompanying increase in wheat trade costs disproportionately affected low and lower-middle-income countries, in particular African nations. Of the ten countries with the largest rise in wheat import costs, five are in Africa, and all are of low or lower-middle income, while of the ten countries with the largest decrease in trade barriers with Ukraine, nine are in Europe (see Figs. S5, S7, and S8 in the SI). Countries such as Nigeria or Angola, while experiencing a similar drop in trade as Norway or France, simultaneously saw an increase in their trade costs. Canadian imports fell by 91%, yet unit trade costs remained nearly constant, while similar drops in Zambia or the DR Congo led to marked increases in unit trade costs, pointing to trading barriers. European countries saw an average 9%

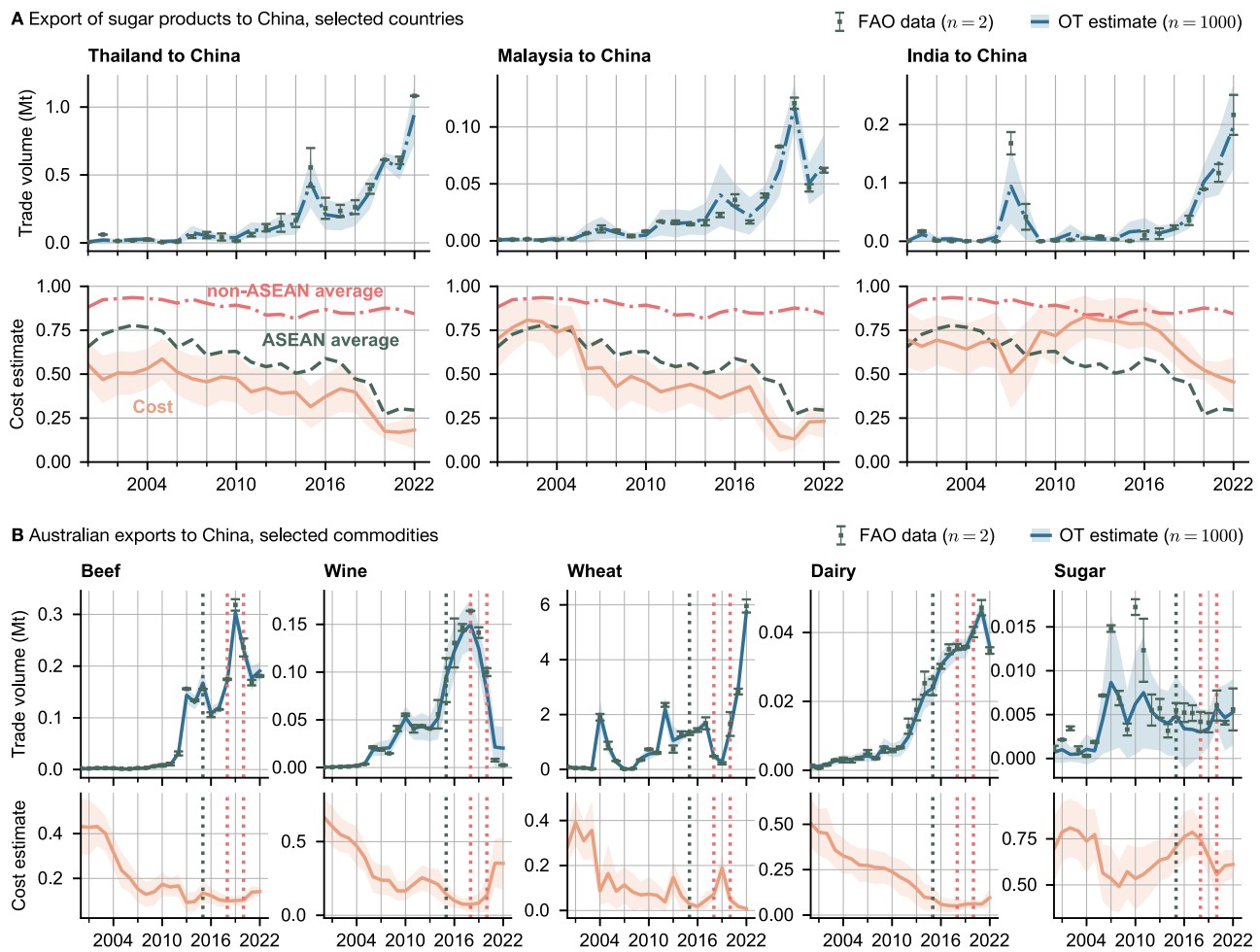

**Fig. 2 | Trade with China. A** Export of sugar products to China from ASEAN (Association of Southeast Asian Nations). Top row: estimated trade volume (light blue) in million metric tons, as well as the reported values. Bottom row: estimated cost, together with the ASEAN and non-ASEAN averages. **B** Australian exports to China, selected commodities. Top row: model estimated flow and FAO data; bottom row: estimated cost. Indicated are the signing of ChAFTA (2015, green dotted line) as well as the start of the US-China and Australia-China trade disputes (2018 and 2020, red dotted lines). Central lines and markers indicate the mean, and errorbands indicate one standard deviation. Sugar products comprise: sugar, refined sugar, syrups, fructose, sugar confectionery. Dairy products comprise: butter, skim milk of cows, cheese, other dairy products.

drop in unit wheat trade costs with Ukraine, while Sub-Saharan Africa saw an average 22% increase (see Fig. S7 in the SI). Imports of wheat from Russia also fell globally (see Fig. S6 in the SI), again affecting Africa particularly severely. European imports of Russian wheat fell by around 74% with an 18% increase in trade costs; African imports fell by an average 80% with a 36% increase in trade costs. While many European countries saw their imports of Ukrainian wheat rise, Russian imports fell sharply. The two largest hubs for Russian wheat, Egypt and Türkiye, saw no change in their import volumes and small declines in their import barriers. Meanwhile, Iran saw a 0.51 point increase in wheat unit trade costs, leading to a 97% percent decline in Ukrainian wheat imports. For Russian wheat, the estimated increase in trade costs was only 0.04, leading to a drop in imports of 53%. Russian-Iranian trade barriers were thus not markedly affected by the war, despite a drop in trade volumes.

## Case study II: Trade in Southeast Asia and Asia-Pacific

A series of free-trade agreements came into effect in Southeast Asia and the Asia-Pacific region in the 2000s and 2010s, significantly among them the China-Australia Free Trade Agreement (ChAFTA) in 2015, the ASEAN-China free trade agreement (ACFTA, gradually entering into force from 2003) and the Comprehensive and Progressive Agreement for Trans-Pacific Partnership (CPTPP) between 11 counties bordering

the Pacific Ocean (2018)[28–30]. Together with China's accession to the WTO in 2001 and its rapid economic growth, these trade agreements coincide with some of the largest increases in trade flows in recent history. In Fig. 2A, we show the trade flow of sugar and sugar products from Thailand, Malaysia, and India to China, as well as the estimated costs. In our model, the cost of importing sugar from Thailand fell consistently from 2000 to 2022, following a general trend for ASEAN countries (bottom row, green line) which commenced around 2005. Indian exports, by comparison, remained relatively low until 2015, when Indian Prime Minister Narendra Modi visited China, and top officials from both sides agreed to increase bilateral trade to US$100 billion by the end of the year. This visit marked a dramatic shift in Indo-Chinese trade, as exemplified by the huge increase in sugar trade. From 2015 to 2022, sugar export cost from India dropped sharply by 33%, precipitating a steep increase in trade. By contrast, trade cost from non-ASEAN members has remained constant over the past twenty years (red line, Fig. 2A).

The PRC is one of Australia's largest export markets for food and agricultural products. Our analysis suggests a precipitous reduction in trade barriers for Australian exports since China's accession to the WTO in 2001 (see Fig. 2B), particularly for beef, wheat, wine, and dairy. Between 2002 and 2010, these commodities saw a 30–50% drop in their respective trade costs. Our estimates indicate that ChAFTA had

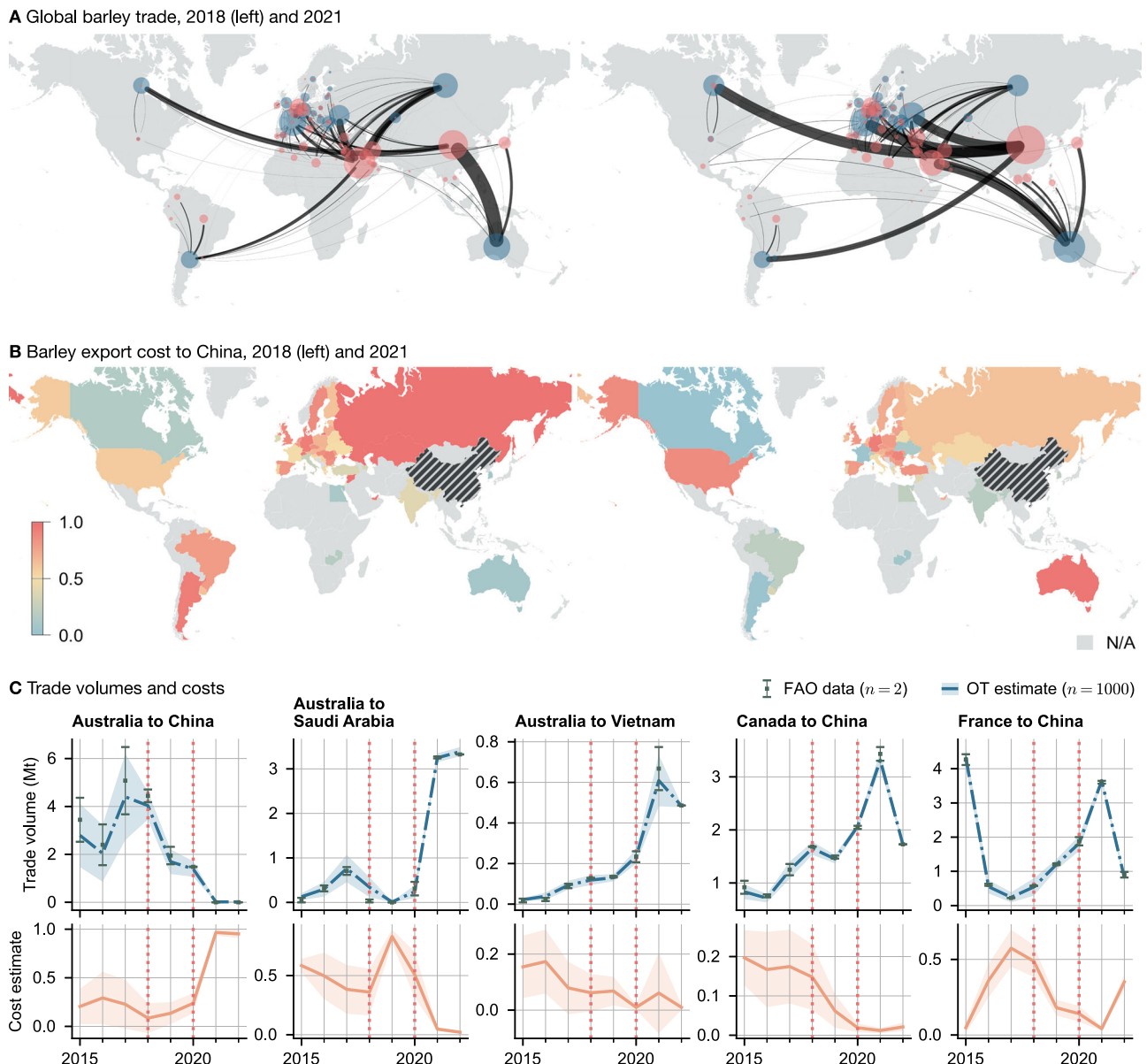

**A** Global barley trade, 2018 (left) and 2021

**B** Barley export cost to China, 2018 (left) and 2021

**C** Trade volumes and costs

FAO data ($n = 2$)   OT estimate ($n = 1000$)

**Fig. 3 | Global barley trade between 2015 and 2022.** After the introduction of Chinese import tariffs on Australian barley in May 2020, the entire supply chain restructured itself, with Chinese demand being supplied from France, Canada, and Ukraine, and Australia increasingly exporting to Saudi Arabia and Southeast Asia. **A** Trade in 2018 and 2021. Importing countries are shown in red, export values in blue, with node sizes and edge widths representing total volumes. **B** Barley export costs to China in 2018 and 2021. **C** Model estimated trade volumes (top row) and cost (bottom row) for selected countries. Dotted lines indicate the start of the US-China and Australia-China trade wars. Central lines and markers indicate the mean, and errorbands indicate one standard deviation.

little effect on Australian trade costs, since it succeeded a period of deepening ties. Dairy trade costs, for instance, had already fallen from 0.51 to 0.1 from 2000 to 2015, thereafter only falling a further 0.04 points until 2020. Wine exports too saw their largest reductions in trade costs between 2000 and 2010, only experiencing a 0.07 drop from 2015 to 2018 compared to the 0.52 point reduction from 2000 to 2015.

In January 2018, the first Trump administration started imposing import tariffs on goods primarily from China. In response, the Chinese government increased tariffs on a variety of products, including agricultural imports. The largest agricultural export from the US to China, soya beans, were targeted by a 25% import tariff[31]. Meanwhile, political tensions between China and Australia caused Beijing to introduce high anti-dumping tariffs on Australian exports such as barley (80.5%) and wine (206%), starting in 2020[32]. Wine trade had previously been tariff-free since the signing of ChAFTA in 2015. Our analysis provides an

estimate of the change in the ease of trading these measures induced (Figs. 2B, 3 and 4). Australian beef, wine and barley imports all experienced large increases in cost, following the implosion of trade volumes. Australia was able to divert some of its excess barley supply to Saudi Arabia, which saw a decrease in trade costs of over 0.8 points between 2019 and 2022 (Fig. 3C). Trade volumes to Vietnam also increased from 200,000 to 800,000 metric tons, though trade costs remained approximately constant. Meanwhile, after 2020 China doubled its barley imports from Canada and France. We found that import barriers from both countries were reduced only slightly in 2021 and rebounded the following year.

**Case study III: Brexit**

In 2016, the United Kingdom voted to leave the European Union, officially exiting the common market and customs union on December 31, 2020. This case study examines the impact of Brexit on British

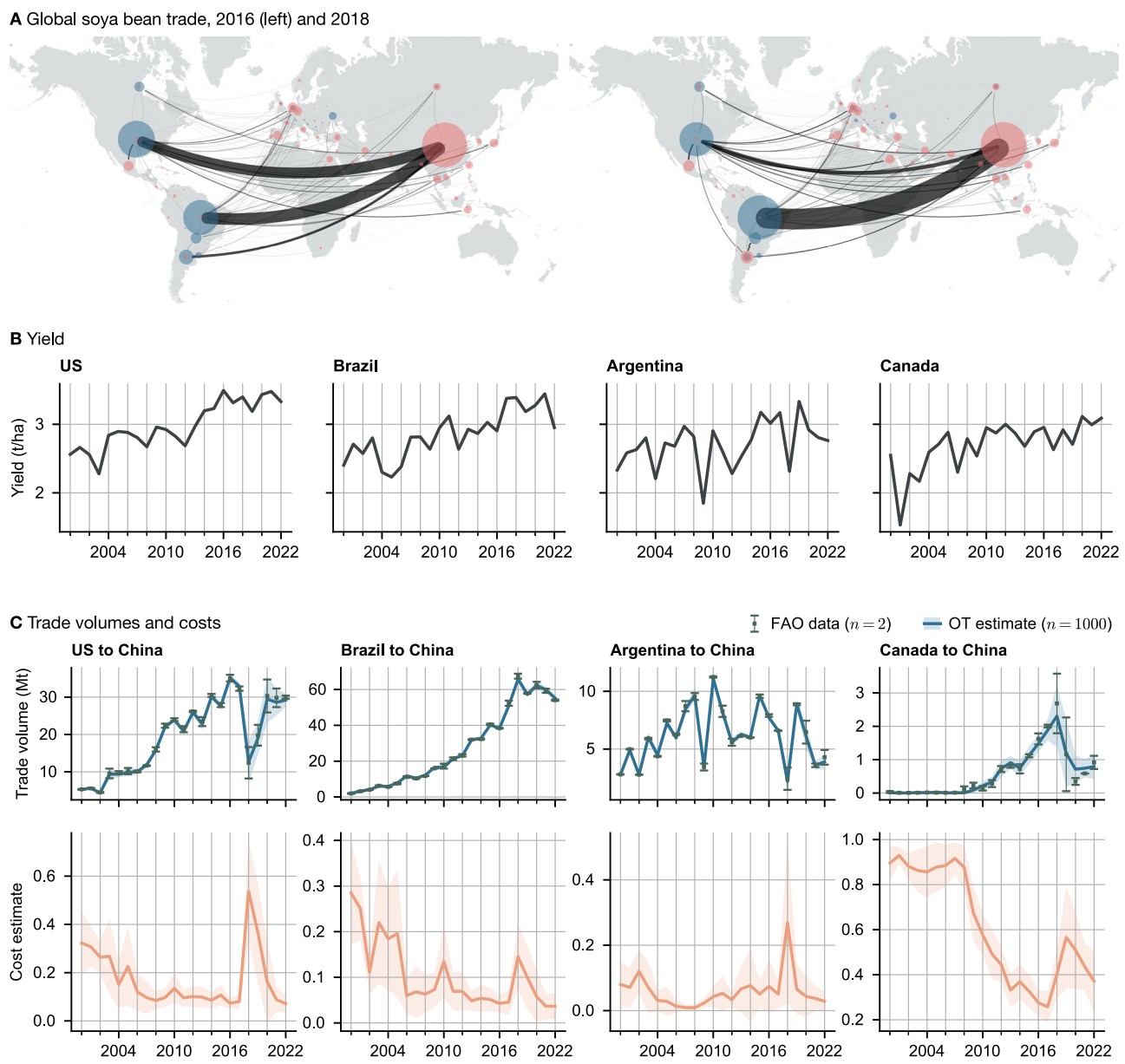

**Fig. 4 | Global soya bean trade. A** In 2018, the Chinese government raised import tariffs on American soya beans in a retaliatory action against US trade restrictions. The shortfall was met by imports from Brazil. **B** Soya bean yield in tons/hectare. Argentina in 2018 experienced a major drop in yields, leading to an increase in exports from the US. **C** Predicted trade volumes in million metric tons (top row) and predicted cost (bottom row). Central lines and markers indicate the mean, and errorbands indicate one standard deviation.

import patterns by comparing vegetable and wine imports from mainland Europe to both the United Kingdom and the Republic of Ireland (ROI), which remains part of the Eurozone and the common market. While both island nations naturally source the majority of their fresh produce from mainland Europe, their trading patterns have evolved in markedly different ways (Fig. 5A). Imports of lettuce from Europe generally fell for the UK, accompanied by a rise in import cost: 25% decrease in trade volume and +0.04 in import costs from the Netherlands, the largest exporter of lettuce and chicory to the UK, as well as a 28% drop in trade from Spain, though with no change in import cost. Ireland increased its imports of lettuce and other greens from the Netherlands, Spain, and Italy, accompanied by a general decrease in trading costs for those products. The ROI's imports of greenery from Portugal fell by 53%, with no change in trade barriers; for the UK, a 71% fall in trade volumes accompanied a 0.12 point increase in trade cost. In the case of the Netherlands, Ireland saw a

consistent reduction in vegetable trade costs, unlike the United Kingdom. It is interesting to note that the United Kingdom sharply increased its imports of vegetables from Morocco, accompanied by a precipitous drop in trade costs, indicating a facilitation of trade between the two countries in the wake of Brexit. This is not true for the ROI: though it increased its imports of Moroccan tomatoes and cucumbers, Irish trade costs remained mostly unchanged.

A clearer-cut trend emerges in the wine trade (Fig. 5B): here, the UK was consistently affected more negatively than the Republic of Ireland: British import costs from all eight countries considered rose by considerably more than those of the ROI. A 9% drop in Spanish wine import by the UK was accompanied by a 7% increase in trading costs, while a 13% drop in Irish imports only meant a 5% increase in costs. Portuguese wine imports to the UK rose by 16%, notwithstanding a 0.05 increase in trade costs. A similar pattern holds for South African, Australian, and New Zealand imports. The EU maintains free trade or

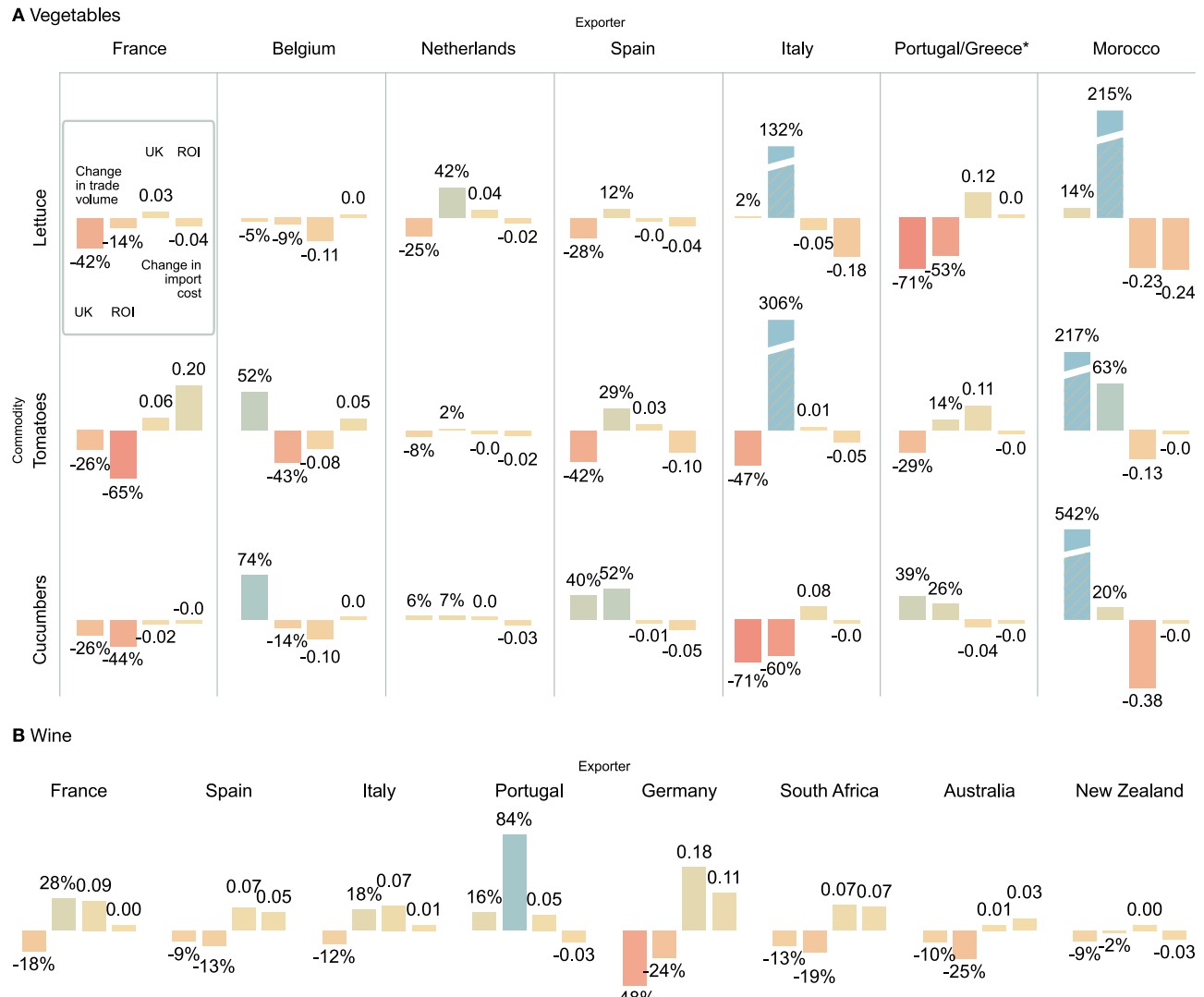

**Fig. 5 | Change in UK and Ireland (ROI) imports, 2016–2022.** For each exporting country, the left two bars indicate the percent change in trade volume between 2016 and 2022 for the UK and the ROI, respectively, the right two bars show the change in import costs. **A** Vegetable imports. Top row: lettuce (including chicory), and other fresh vegetables; middle row: tomatoes; bottom row: cucumbers and gherkins. *Note that the exporter for cucumbers is Greece, for Lettuce and Tomatoes it is Portugal. **B** Wine imports.

regulatory agreements removing wine import duties with the former two[33,34]. When the UK left the European Union, wine from Australia entered the UK at the Global Tariff rate, which in mid-2023 was eliminated under the Australia-United Kingdom FTA[35]. South African wines, by contrast, continued to be imported to the UK tariff-free post-Brexit[36]. Yet here too, the United Kingdom's 13% decrease in imports was driven by a 0.07 increase in trading costs; Ireland, by contrast, imported 19% less wine, driven by a comparable 0.07 point increase in trading costs.

## Comparison with gravity model

Lastly, we compare the performance of our method with a traditional gravity model[11,37–39], as specified in Eqs. (1) and (2). The covariates include geographic distance, shared borders, colonial ties, common language, regional trade agreements, tariffs, and importer/exporter fixed effects. We estimate the coefficients using Poisson Pseudo Maximum Likelihood estimation[22,40] and compare the accuracy of the estimated transport plans **T**. Figure 6A, B shows scatter plots of the OT (blue) and the gravity (orange) estimates against the FAO data. For all commodities studied, a linear fit through the OT estimates yields a

near-perfect slope of $m = 1$ with a Pearson coefficient close to 1, particularly providing an exact fit for the upper tail of the trade value distribution. In contrast, the gravity model's performance is much more volatile, with linear fits ranging from a Pearson coefficient of between 0.975 (best) to 0.699 (worst) (see also SI). Due to model misspecification, the fits to the tails of the distributions are generally poorer. Consequently, Fig. 6C suggests that the OT approach clearly outperforms the gravity model in terms of RMSE, often by an order of magnitude. Figure 6D illustrates that OT estimates typically fall within one standard deviation of the data uncertainty, whereas gravity estimates tend to range from one to two, at times even three to four standard deviations. The gravity model also exhibits much higher variance in accuracy compared to OT. We further investigate an alternative specification by replacing time-varying country-level and time-invariant pair regressors with exporter-time, importer-time, and exporter-importer fixed effects in a three-way gravity model[41], capturing multilateral resistance terms more effectively (see Eq. (2) and Fig. S9 in the SI for details). As this formulation incorporates higher degrees of freedom in the three-way specification, its performance naturally converges toward that of OT, which more accurately fits high-

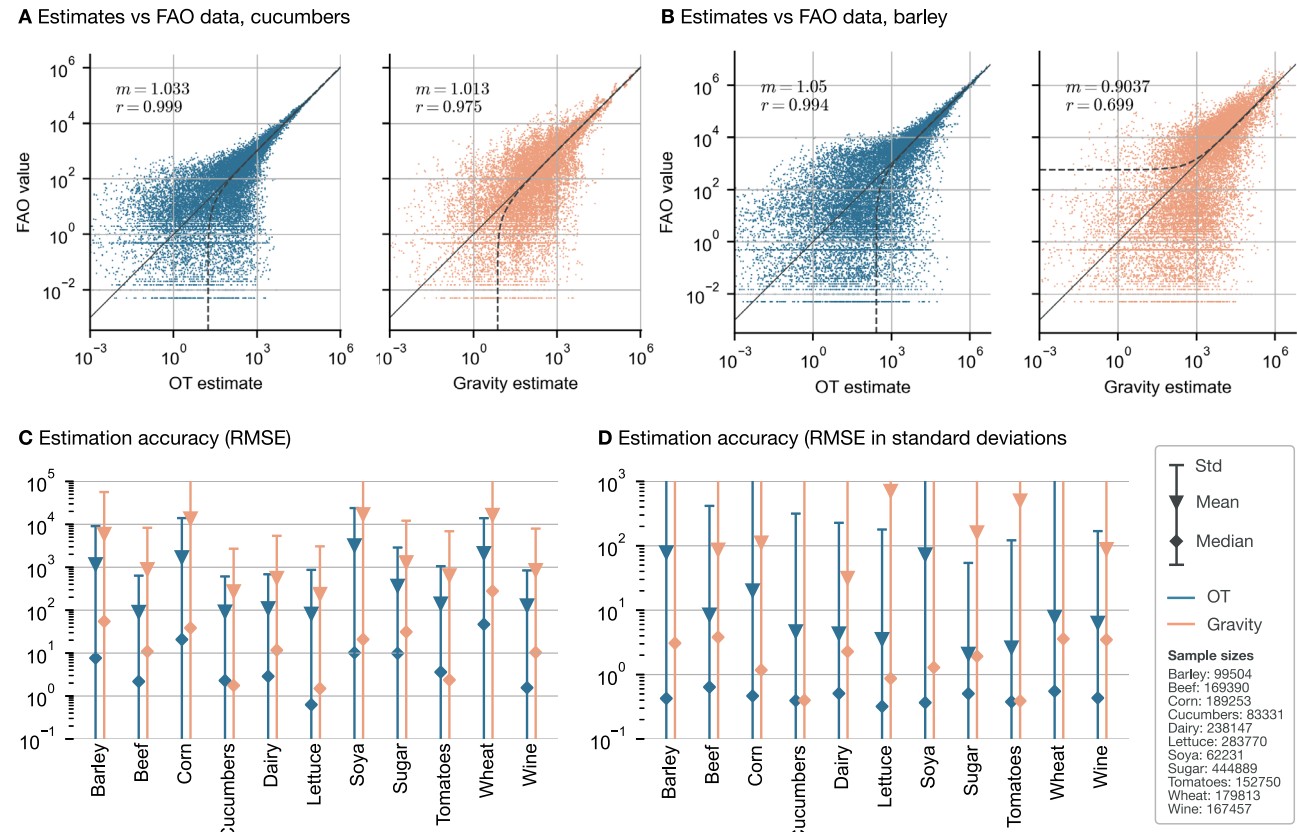

**Fig. 6 | Comparison with gravity model. A**, **B** Comparison plot of the OT and gravity estimates (*x*-axis) versus the true data (*y*-axis) on two selected commodities. Also shown is a linear fit (dotted line), its estimated slope *m*, the Pearson coefficient of the fit *r*, and the line *y* = *x* (solid line). See Fig. S10 in the appendix for an overview of all commodities. **C**, **D** Comparison of the RMSE accuracies of the estimated transport volumes of the OT approach (blue) and the gravity model (orange). Values are averaged over all countries and years, with the errorbars showing one standard deviation from the mean (triangular marker). Also shown are the median values (diamond markers). Shown are the RMSE (**C**) and the RMSE in units of the standard deviation on the true data (**D**).

value trade flows. Incorporating time-varying bilateral trade regressors would likely further narrow this gap, reinforcing the validity of the OT approach. Notably, OT provides trade cost estimates even without such regressors—many of which are difficult to quantify.

## Discussion

This paper introduces a versatile approach for identifying the drivers and barriers of global commodity trades. Using OT theory, we are able to obtain a cost structure that is more expressive than a covariate-based gravity approach. Our estimates are thus orders of magnitude more accurate than traditional gravity models, matching the performance of high-dimensional fixed-effects specifications while maintaining consistent accuracy across datasets. The optimal transport approach models trade networks as a dynamical, interconnected system, allowing to capture complex rearrangements and network response dynamics to e.g. trade wars, conflicts, or shifts in political relations. Though the current work looks only at global agrifood markets, the methodology proposed is general and applicable to commodity flows, financial markets, or banking networks[42]. Beyond economics, the OT approach also relates e.g. to global migration flows, which can be estimated from migrant stock data[43–45]. Future work could explore correlations between related commodities within this framework, as well as develop hybrid models that combine observed covariates with data-driven residuals via semi-structured OT costs. Another promising direction is counterfactual analysis using the conditional equilibrium framework of Yotov et al.[11]. Extending the model to a full general equilibrium setting with endogenous production and pricing is also a natural next step.

## Methods

### Entropy-regularised optimal transport

In OT, one wishes to find the optimal flow of mass from a source distribution to a target distribution, while minimising an overall transport cost. This abstract problem has a wide range of applications in economics, logistics, image restoration, transport systems, or urban structure[21,46,47].

Consider an *m*-dimensional space *X*, an *n*-dimensional space *Y*, and **C** a measure on *X* × *Y*. The entries of **C** correspond to the cost of transporting mass from one location in *X* to a target in *Y*. Given two probability measures $\boldsymbol{\mu} \in P(X)$ and $\boldsymbol{v} \in P(Y)$ (the supply and demand), the OT problem consists in finding a transport plan **T** minimising the overall cost Eq. (3). The transport plan **T** must also satisfy the marginal constraints

$$\sum_j T_{ij} = \boldsymbol{\mu} \text{ and } \sum_i T_{ij} = \boldsymbol{v}. \qquad (9)$$

In practice one usually considers the entropy-regularised OT formulation, which can be solved much more efficiently[48]; here, an additional term is added to the objective:

$$\min_{\mathbf{T}} \sum_{ij} C_{ij} T_{ij} + \varepsilon \sum_{ij} T_{ij} \left( \log T_{ij} - 1 \right), \qquad (10)$$

where $\varepsilon > 0$ is a positive regularisation parameter. This regularisation prevents monopolisation, i.e. demand being supplied from only a few sources.

The constrained optimisation problem Eq. (10) can be solved by considering the Lagrangian

$$\mathcal{L} = \sum_{ij} T_{ij} C_{ij} + \left\langle \boldsymbol{\lambda}, \sum_j T_{ij} - \boldsymbol{\mu} \right\rangle + \left\langle \boldsymbol{\eta}, \sum_i T_{ij} - \boldsymbol{\nu} \right\rangle$$
$$+ \varepsilon \sum_{ij} T_{ij} \left( \log T_{ij} - 1 \right), \tag{11}$$

with $\boldsymbol{\lambda} \in \mathbb{R}^m$ and $\boldsymbol{\eta} \in \mathbb{R}^n$ Lagrangian multipliers. Minimising $\mathcal{L}$ with respect to $\mathbf{T}$ gives the solution

$$T_{ij} = e^{-\lambda_i/\varepsilon} e^{-C_{ij}/\varepsilon} e^{-\eta_j/\varepsilon} \tag{12}$$

or

$$\mathbf{T} = \boldsymbol{\pi} e^{-\mathbf{C}/\varepsilon} \boldsymbol{\Omega}, \tag{13}$$

where $\boldsymbol{\pi} = \mathrm{diag}(e^{-\lambda_1/\varepsilon}, \cdots, e^{-\lambda_m/\varepsilon}) \in \mathbb{R}^{m \times m}$ and $\boldsymbol{\Omega} = \mathrm{diag}(e^{-\eta_1/\varepsilon}, \cdots, e^{-\eta_n/\varepsilon}) \in \mathbb{R}^{n \times n}$ are diagonal matrices of Lagrangian multipliers.

Finding $\boldsymbol{\Pi}$ and $\boldsymbol{\Omega}$ is achieved through an iterative scaling procedure that is variously called Iterative Proportional Fitting (IPF), RAS, or Sinkhorn's algorithm[48–50]. Define $\mathbf{M} = e^{-\mathbf{C}/\varepsilon}$; then, given an initial guess $\boldsymbol{\Pi}^0$, we update $\boldsymbol{\Omega}$ to satisfy the first marginal constraint eq. (9)

$$\boldsymbol{\Omega} \mathbf{M}^\top \boldsymbol{\pi}^0 = \boldsymbol{\nu}. \tag{14}$$

Solving for $\boldsymbol{\Omega}$ gives

$$\boldsymbol{\Omega}^0 = \frac{\boldsymbol{\nu}}{\mathbf{M}^\top \boldsymbol{\pi}^0} \tag{15}$$

where the division is understood element-wise. Similarly, we obtain the next update for $\boldsymbol{\Pi}$ as

$$\boldsymbol{\pi}^1 = \frac{\boldsymbol{\mu}}{\mathbf{M} \boldsymbol{\Omega}^0}, \tag{16}$$

and so on. The algorithm can thus be summarised as follows:

**Algorithm 1. Sinkhorn's Algorithm**
1: **Inputs:**
$\mathbf{M}$ (Exponential of cost matrix)
$\boldsymbol{\mu}, \boldsymbol{\nu}$ (marginals)
2: Initialise the first Lagrangian multiplier $\boldsymbol{\Pi}^0$
3: **for** $n$ iterations **do**
4: $\boldsymbol{\Omega}^{i+1} \leftarrow \frac{\boldsymbol{\nu}}{\mathbf{M} \boldsymbol{\pi}^i}$
5: $\boldsymbol{\pi}^{i+1} \leftarrow \frac{\boldsymbol{\mu}}{\mathbf{M}^\top \boldsymbol{\Omega}^{i+1}}$
6: **end for**

Under certain conditions, convergence of the algorithm to a unique solution is guaranteed[51,52].

The classic OT problem Eqs. (3) and (4) can be interpreted as the central planner's problem of finding the optimal assignment or matching of supply and demand. The entropy-regularised OT problem can be viewed as a similar optimal assignment problem, but subject to uncertainty and/or randomisation. The dual problem to entropy-regularised OT, i.e. minimising (5) subject to (3), is given by

$$\max_{\boldsymbol{f}, \boldsymbol{g}} \langle \boldsymbol{f}, \boldsymbol{\mu} \rangle + \langle \boldsymbol{g}, \boldsymbol{\nu} \rangle - \varepsilon \sum_{i,j} e^{\frac{f_j + g_i - C_{ij}}{\varepsilon}}. \tag{17}$$

In the limit $\varepsilon \to 0$ the last term ensures that the dual potentials $\boldsymbol{f}$ and $\boldsymbol{g}$ satisfy

$$\boldsymbol{f} \oplus \boldsymbol{g} \leq \mathbf{C} \tag{18}$$

where $\boldsymbol{f} \oplus \boldsymbol{g} = \boldsymbol{f} \mathbf{1}_m^T + \boldsymbol{g} \mathbf{1}_n^T$. Condition (18) corresponds to the admissibility condition of the dual non-regularised OT problem. In this context $\boldsymbol{f}$ and $\boldsymbol{g}$ can be interpreted as the minimal cost of picking up and dropping off a good at locations respectively. The problem of finding the optimal plan $\mathbf{T}$ is thus split into determining the optimal cost of collecting and delivering goods. The constraint (18) ensures optimality: if $f_i + g_j > C_{ij}$, that is, the cost of picking up a good at location $i$ and dropping it off at location $j$ is larger than the transportation cost, it cannot be optimal.

### Neural inverse optimal transport

In inverse OT one wishes to infer the underlying cost $\mathbf{C}$ from (partial) observations of transport plans $\mathbf{T}$, which are usually solutions to entropy regularised OT problems. Rewriting Eq. (13), we have

$$\mathbf{C} = \varepsilon \left( \log \boldsymbol{\pi} + \log \boldsymbol{\Omega}^\top - \log \mathbf{T} \right);$$

thus $\mathbf{C}$ is only determined up to an additive decomposition into row and column vectors, with any transformation of the kind

$$C_{ij} \mapsto C_{ij} + \alpha_i + \beta_j$$

leaving the transport plan $\mathbf{T}$ invariant, since the transformation can be absorbed by the scaling vectors. To constrain the problem, we can bound the cost $C_{ij} \in [0, C_{max}]$, and demand that the inverse of 0 should be $C_{max}$, i.e.

$$u : \mathbf{T} \mapsto \mathbf{C}, \quad u(0) = C_{max}.$$

This is a natural restriction, since it implies that the cost on edges with zero transport flow should be maximal. If this maximum is attained in every row and every column of the cost matrix, the row- and column-shifts $\alpha_i$ and $\beta_j$ must satisfy

$$C_{max} = \max_j C_{ij} \overset{!}{=} \max_j (C_{ij} + \alpha_i) \leq C_{max} \ \forall \ i \Rightarrow \alpha_i = 0.$$

(and similarly $\beta_j$). Thus, under these conditions the cost matrix is uniquely determined (see also Fig. S1 in the SI).

To infer the cost matrix function $\mathbf{C}(t)$ from a dataset of transport plan observations $\mathbf{T}(t)$, we build on the neural parameter estimation method first introduced in ref. 23 and subsequently expanded upon in ref. 24. We wish to train a neural network $u$ to solve the inverse OT problem $\mathbf{C}(t) = u(\mathbf{T}(t))$. We do so by constructing a loss function based on the OT equations, i.e.

$$J = \| \widehat{\mathbf{T}}(\widehat{\mathbf{C}}) - \mathbf{T} \|_2^2 + \sum_{(i,j) \in \mathcal{S}} (C_{ij} - C_{max})^2. \tag{19}$$

Here, $\widehat{\mathbf{T}}(\widehat{\mathbf{C}})$ is the estimated transport plan obtained by solving Sinkhorn's algorithm alg.[1] until convergence (determined by a numerical tolerance criterion), and

$$\mathcal{S} := \{(i,j) \mid T_{ij} = 0\}$$

are the zero-flow edges of $\mathbf{T}$. The second term thus enforces the maximum value $C_{max}$ to be attained on $\mathcal{S}$. Crucially, the solution of entropy-regularised OT is differentiable with respect to its inputs, and the derivative of $\widehat{\mathbf{T}}(\widehat{\mathbf{C}})$ with respect to $\widehat{\mathbf{C}}$ can be computed numerically.

Thus, the loss $J$ can be minimised using gradient descent methods. While the dependence of the stopping time on $\mathbf{C}$ introduces potential non-differentiabilities at iteration-change boundaries, in practice these events are rare and did not cause instability in our experiments; the issue can also be avoided entirely by instead fixing the number of iterations at a sufficiently large value. The data is processed in batches, and a gradient descent step performed on the neural network parameters after each batch. The loss is only calculated for links with trade flow >0. Note that our goal is not to predict future trade-flows, but rather to infer an underlying cost which drives the flows subject to the OT model. We therefore do not require large volumes of data, as would be typical in a prediction task.

As mentioned, the FAO dataset contains two values for each entry $T_{ij}$: one reported by the exporter, and one by the importer. Let $\mathbf{T}^E$ be the transport plan where all entries are those reported by the exporters, and $\mathbf{T}^I$ those where all are reported by the importers. The training data—i.e., the data we use to train the function $u$—consists of only these two transport plans for each year: $\{\mathbf{T}^E(t), \mathbf{T}^I(t)\}$, giving a total training set size of $2 \times L$, where $L = 23$ are the number of observation points. A hyperparameter sweep showed that using a deep neural network with 5 layers, 60 nodes per layer, and hyperbolic tangent activation functions on all layers but the last, where we use a sigmoid, gives best results. Using a sigmoid activation function on the last layer ensures $0 \le C_{ij} \le 1 = C_{max}$. We use the Adam optimiser[53] to train the neural network. We pool all FAO trade matrices to only contain those countries that account for 99% of import and export volumes, subsuming all other countries in an 'Other' category (thereby ensuring that no flow is lost). Entries for which neither the importer nor the exporter have reported a value are assumed to be zero, and we constrain the cost matrix to be maximal on these entries. Entries for which only the exporter or the importer have reported a value (but not the other) are presumed missing in the respective table, and are masked in the loss function. With this approach, on average about 20% of entries are masked in the transport plan (see Fig. S2 in the SI). Entries for which all reported values are missing populate the zero-flow edge set $\mathcal{S}$.

### Uncertainty quantification
Uncertainty on the estimates stems from two sources: one, the degree to which the minimiser of the inverse problem (19) is ill-defined (i.e. the number of possible cost matrices that all fit the problem equally well), and two, the uncertainty on the transport plans themselves. To address the first, we use an ensemble training approach[23,24] and train a family $\{u_k\}$ of 10 neural networks for each commodity in parallel. Even though the inverse problem is theoretically well-posed when the forward problem (10) admits a solution, in the case of the FAO data we are solving a minimisation problem. To incorporate the uncertainty on the transport plans, we pass random samples of $\mathbf{T}$ through each of the trained neural networks $u_k$. These samples are obtained by selecting either $T_{ij}^E$ or $T_{ij}^I$ uniformly at random for each entry of the transport plan, and passing this sample through each neural network. Repeating this $n$ times gives $n$ samples of $\mathbf{C}$, and inserting each estimate of $\mathbf{C}$ into Sinkhorn's algorithm gives $n$ estimated transport plans $\hat{T}$. We generate a total of $n = 100$ samples for each year and neural network (see also Fig. S4 in the SI). The uncertainty estimates obtained by our method then provide an indication of how strongly a given set of trade flows informs the underlying cost.

### Reporting summary
Further information on research design is available in the Nature Portfolio Reporting Summary linked to this article.

### Data availability
The data generated in this work, as well as the trained neural networks, are available in Huggingface repository 10.57967/hf/7190 under accession code https://huggingface.co/datasets/ThGaskin/OT_Trade.

### Code availability
The code used to train the neural networks and produce the analysis results is available at https://github.com/ThGaskin/inverse-optimal-transport[54]. Instructions for running the model are given in the README.

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

## Acknowledgements

T.G., A.D., and M.T.W. acknowledge partial support by the EPSRC grant EP/X010503/1. A.D. also acknowledges support of a Royal Society APEX award (APX/R1/180133), under which this work was initiated. G.D. acknowledges support by the Ministry of Education of the Republic of Korea and the National Research Foundation of Korea (NRF-2024S1A3A2A07046144).

## Author contributions

T.G., G.D., M.T.W., and A.D. designed the research and wrote the paper. T.G. and G.D. performed the numerical experiments.

## Competing interests

The authors declare no competing interests.
