## [Transparent Peer Review file · Nature Communications]

Modelling Global Trade with Optimal Transport

Corresponding Author: Dr Thomas Gaskin

Version 0:

Reviewer comments:

Reviewer #1

(Remarks to the Author)

A. Overall Assessment

Let me thank the Authors for this very interesting piece of work. As an economist working with gravity models I was not that familiar with the rich literature linking issues in trade and optimal transport, and I find it very interesting. Also, the three different case studies which highlight different episodes and aspects of the complexities of global trade and different testbeds for the methodology are very much needed. I will also mention that I have little experience with working with Neural Networks (henceforth NNs), so I will mainly stick in this report to the overall gist of the paper and on the section regarding gravity models. I find the paper in general quite clear, well written and well structured. I think there are however some major and minor issues that I believe should be addressed.

B. Major Comments

1) There are some features of the NN methodology that should be made clearer. For example, the Authors do not clarify whether the data the model is trained on are nominal flows of trade for all years in the sample (panel structure) or a single (or a sequence thereof) cross section, i.e. a single year. The confusion stems from calling "T" both the matrix of "transport plans" and that of trade flows. Although, the two might be the same in practice (what one observes), they are very different conceptually.

2) Could one try to perform exercises similar to what is done in the paper using the marginals only? This is a well-known issue in the Input-Output literature and often indeed solved using variants of the IPFs and the RAS algorithm. Would be curious to know if there may be some space for this.

3) I find some of the descriptions of Case study I a bit confusing: when trade is mentioned, it is implicitly describing wheat exports only? Without explicitly mentioning this, one might think that change in "trade volumes" is about total trade and not only wheat.

4) B figures may also require a bit more elaboration: one is struck by the huge increase for some countries, but this is not discussed in the text. This also relates to the fact that it looks as if the change in trade cost is only relevant for increases in cost (but not decreases). Is this correct?

5) I would maybe suggest the authors to corroborate the results on the C figures with simple Spearman correlation coefficients to understand the underlying relationship between change in cost and change in trade.

6) Gravity model (GM):

6.1 I find the comparison with the gravity model a bit unsatisfactory as it is now. For example, Eq. (20) does not make sense as it is currently formulated: the variables O_i , E_j , χ_j would be absorbed into the fixed effects. Since Table S1 reports however their coefficients (which would not be defined if the specification was in the current form) I guess that the authors are really specifying one model for each good, i.e. $\log T_{ijl}$ where l is the product.

6.2 To verify this I also went to the Git page the authors provide but did not find the results - I might have missed this, but if

not I would advise the authors to provide reproducible codes also for the empirical specification. In general, I think this should be stated more explicit in the text as well as correct the equation.

6.3 The natural question would then be: why shouldn't the comparison with the GM be done also with goods fixed effects (either good specific γ_i or interacted with origin and destination). I would think the latter to be a fairer comparison due to the stringency of the NNs approach, cf. the discussion on multilateral resistance terms in the gravity literature.

6.4 A similar argument applies also to the time dimension. First, is not clear whether the comparison is done on a single year (cross-section) or for all years together (pooled sample). Then, if the NN is trained using all the data, then also in this case I believe a solid comparison would require the inclusion of time fixed effects.

6.5 A follow-up point from the previous one would be also a comparison in terms of the number of parameters of the two models. Indeed, one drawback of adding more fixed effects in the specification is that one should then estimate a lot of parameters. This would of course give a much better fit and less degrees of freedom. How would the GM and the NN compare in terms of this? This should be a further dimension to discuss for a fairer comparison between the two approaches.

C. Minor Comments

- The B images in Case Study 1 would require a different scale
- I believe the reference 22 is more subtle than what reported - aggregation matters when parameters vary across sectors (akin to the case where good specific fixed effects are included)
- I would also suggest the authors to move one of the figures (perhaps Fig 2) of the case studies in the appendix. I think there is some graphical redundancy and one has to do a lot of back and forth between text and images.

(Remarks on code availability)

In order to verify some points that were not clear upon reading the manuscript, I went to the Git page the authors provide but did not find some results. I would advise the authors to provide reproducible codes for the empirical specification as well.

Reviewer #2

(Remarks to the Author)

(Remarks on code availability)

Reviewer #3

(Remarks to the Author)

This paper uses an optimal transport model in combination with a deep neural network to get a time varying function of trade costs from noisy trade volumes while demand and supply constraints are met. The method is applied to global food and agricultural data. The authors demonstrate that the performance of the proposed method to shocks such as Brexit, the Russian invasion in Ukraine and trade disputes with China. The paper is overall well written and the analysis is well explained.

I have two main concerns. First, to my understanding the authors take as given the demand and supply (parameter μ and ν) and solve for the optimal trade flows subject to these amounts. If that's right, then demand and supply are not endogenous and we are missing important equilibrium effects. This is particularly important for performing counterfactuals.

Second, and somewhat related, although I take the authors' point that gravity models are quite stylized by parameterizing trade costs by a handful of variables, I am not sure full flexibility is desirable. The reason is the following: allowing for a fully flexible matrix of trade costs makes me worry about the validity of this specification in counterfactuals. In other words, it is less convincing that these trade costs will be invariant in counterfactual predictions, especially if the counterfactual is "far from our data".

Moreover, it was not clear to me how the authors performed the empirical analysis. It would be helpful if the authors explained this a little better. For instance, related to the above: it is reassuring that the proposed methodology can better predict what happened after the specific shocks studied; however, it seems to me the authors use data that include these shocks. It would be better to do this as an out of model fit type of exercise.

Other comments:

1. The regularization converts the standard linear LP optimal transport problem into a nonlinear one. The resulting maximum entropy approach and its economic meaning must be further explained considering the role of the hyperparameter ϵ , which needs further elaboration in connection with the case studies also.

2. The duality in (17) and (18) refers to the LP version. The authors should explain the dual version of the nonlinear regularized version.

3. Training is based on random batches of T which produce 'target' cost values. The authors should elaborate further the behavior outside the training set over a test set.

(Remarks on code availability)

Reviewer #4

(Remarks to the Author)

Report for "Modelling Global Trade with Optimal Transport"

Summary

This paper proposes inverse optimal transport as a method to infer trading cost functions between countries. The main assumption is that the observable trading volumes (as a function of time) are entropy-regularised optimal transport plans with an underlying time-dependent cost function (measuring a general cost of transporting commodities from a country to another). It is shown that compared to classical gravity models, this approach gives more accurate reconstructions of the trading volumes. Moreover, the inferred cost functions can be used to study the effect of extrinsic factors (wars, tariffs) on trade patterns that cannot be easily uncovered from the trade volumes alone. The approach is demonstrated in several case studies using global trade matrix datasets from FAO.

Comments

The paper is well written and the approach is simple and elegant. I have several questions about the mathematical aspects of the methodology that the authors may wish to clarify.

1. The primary approach in this paper is an inverse optimal transport problem: given an observation of the transport plan, one needs to recover the transport cost function. Can the authors comment on the well-posedness of this inverse problem? In particular, is it generally true that there exists a unique C such that the optimal transport plan is equal to an observed T ? How robust is this with respect to missing data in T ? Since this is the central idea behind the algorithm, it is advantageous to discuss this at length.

2. The parameter ϵ in equation (10) appears to be a hyper-parameter. How does ϵ affect the well-posedness of the inverse problem, and more importantly, how does the inferred C depend on ϵ ? The authors may wish to demonstrate some robustness of their results with respect to the choice of ϵ , or discuss how this parameter should be chosen in a reasonable way. Similarly and perhaps less importantly, the number of iterations of the sinkhorn algorithm in equation (19) (definition of \hat{T}) is also a hyper-parameter. Please also discuss its effect and associated robustness.

3. Figure 7 compares the performance of the OT approach with the gravity model. I have some questions about this comparison:

- The improvement is on the reconstruction of the observed trade volumes (transport plan). For the gravity model, a linear model is used on the cost C (which then determines T). One would expect that with a greater number of well chosen covariates in the regression below equation (1), one may get better reconstruction for the gravity model. Have the author ruled out that it is due to the small number of (or badly chosen) covariates that cause the reconstruction performance of the gravity model to be sub-optimal?

- The results here are not a demonstration of whether the inferred C is more accurate. Do the authors have any way of demonstrating that the inferred C is correct, perhaps on a synthetic example?

- It is claimed at the end of page 3 that uncertainty quantification performance of the OT approach is also superior. Where is this demonstrated?

4. Does the current approach train a different cost C for each traded commodity? If so, does it make sense to account for correlations on C for different commodities instead?

Minor issues

1. After Equation 5: you may wish to cross reference the definition of \mathcal{H} in methods, since it is the negative entropy, not the entropy - this may confuse some readers.

2. Equation 8: the notation here is rather confusing as u is a function mapping transport maps to costs as defined in equation 7. If the LHS is a probability density function over costs, then the RHS should be something like $\rho(u^{-1}(C)) \det \frac{\partial u^{-1}}{\partial T}(C)$?

3. End of page 3: "both prediction accuracy and uncertainty", do you mean "uncertainty quantification"?

4. In the analysis on case studies the trading cost changes are reported in "units". Can the authors explain what units are these, and how one should interpret its magnitude?

(Remarks on code availability)

Version 1:

Reviewer comments:

Reviewer #1

(Remarks to the Author)

The Authors have satisfactorily replied to all my comments and remarks. I suggest publication after a careful and further check of typos and grammar is performed.

(Remarks on code availability)

The updated code seems correct.

Reviewer #2

(Remarks to the Author)

(Remarks on code availability)

Reviewer #4

(Remarks to the Author)

The authors have addressed my comments sufficiently. I think the more careful well-posedness and robust analysis has improved the algorithm (and its presentation).

I have one additional comment in view of the revision: The authors clarified that the sinkhorn algorithm is in fact not run for a fixed number of iterations but run till a tolerance is reached. In this case, the number of sinkhorn iterations would also be a (discrete) function of C . This would pose some issues when you back-propagate to compute the derivative of \hat{T} with respect to C , as the function is not differentiable at points at which the number of iterations jump. This is probably a minor in terms of numerics, but it would be good to be careful at least in the explanation of your approach.

(Remarks on code availability)

I have attempted to go through the code and have some suggestions.

1. host the data folder at some hosting site (e.g. huggingface, Harvard dataverse, etc) so that one is not forced to download the data when cloning the repository (it's a few GB).

2. The requirements is not complete, e.g. geopandas is missing, netcdf4 is missing (both throw errors in the notebook - and I cannot seem to load the dataset in cell 4, may have something to do with git ifs). I suggest the author double check the repository using a clean environment and see if all the steps can be reproduced successfully.

Version 2:

Reviewer comments:

Reviewer #4

(Remarks to the Author)

The raised issues were resolved. I am happy to recommend acceptance of the paper.

(Remarks on code availability)

Modelling Global Trade with Optimal Transport

Response to reviewers

We would like to thank the reviewers for their thoughtful and constructive feedback on our manuscript. The comments were extremely helpful and have led to substantial improvements in both the methodology and presentation of the work. In particular, Reviewer 4's observations regarding the well-posedness of the inverse problem prompted a redesign of our loss function and training procedure. We have now adopted an ensemble-based approach, training multiple neural networks in parallel. This allows us to capture a more expressive form of uncertainty in the estimates, reflecting the diversity of cost functions that can explain the observed trade dynamics. While the overall results remain qualitatively and quantitatively similar, the approach is now better grounded theoretically. We also realized that our previous description of the gravity model may have been misleading. Although the original implementation already aligned with Reviewer 1's suggestions, we have revised the text to make this clearer.

Attached is a point-by-point response to the reviewer comments. Changes to the manuscript are highlighted in green; where an entire section has been added, its heading appears in green for ease of reference. We have also added several new figures to the supplementary material. We believe the reviewers' insights have led to a significantly improved manuscript, and are grateful for the opportunity to revise it.

We look forward to hearing from you,

Yours sincerely,

Thomas Gaskin, Guven Demirel, Marie-Therese Wolfram, Andrew Duncan
16 July 2025

Reviewer 1

Major Comments

1. *The authors do not clarify whether the data the model is trained on are nominal flows of trade for all years in the sample (panel structure) or a single (or a sequence thereof) cross section, i.e. a single year. The confusion stems from calling \mathbf{T} both the matrix of “transport plans” and that of trade flows. Although the two might be the same in practice (what one observes), they are very different conceptually.* We thank the reviewer for highlighting the difference between trade flows (observations) and underlying transport plans that generate instances of flows. For clarity, we denote the transport plans by \mathbf{T} and the matrix of observed flows by \mathbf{T}^E (\mathbf{T}^I), as reported by the exporting (importing) country. The training data consists of the full time-series of observed flows for each commodity, using \mathbf{T}^E and \mathbf{T}^I values for L time-points (23 years). The input to the neural network is a transport plan observation for a *single year*, i.e. $\mathbf{T}(t)$; see equation 7 in the manuscript.
2. *Could one perform similar exercises using the marginals only?* This approach can, in principle, be generalised to problems where one observes marginals only. This problem is, however, significantly more ill-posed than the problem investigated in this paper, since infinitely many transportation plans lead to the same marginals. It would thus require stronger (and possibly more application-specific) assumptions on the underlying problem. See also response to Reviewer 4, Q1.
3. *When trade is mentioned, is it referring only to trade of a single commodity? Without explicit mention, one might think that change in “trade volumes” refers to total trade and not only wheat.* Indeed—trade volumes in our article *always* refer to a single commodity. We have changed the wording in some places to this effect to prevent confusion.
4. *B figures require more elaboration: one is struck by the huge increase for some countries, but this is not discussed in the text. This also relates to the fact that it looks as if the change in trade cost is only relevant for increases in cost (but not decreases). Is this correct?* The colourbar scale was previously incorrect, as we had accidentally multiplied the figures by 100 twice (to represent percentages). It has now been corrected. Nevertheless, a small number of countries did see a very large increase in wheat trade. These countries are essentially all located in Eastern Europe; India (which also experienced a large increase in relative terms) by our estimates imported 0.1 tonnes of wheat in 2020, and 4.8 tonnes of wheat in 2021—technically a large relative increase, but small in absolute terms when compared to the Polish increase of 525,000 tonnes. We have added a statement to this effect to the figure caption.

Regarding the question of whether change in trade cost is only relevant for increases in trade volume—this is not so, as can be seen in the newly added figure S7 in the appendix. A positive change in cost (y-axis) is generally associated with a drop in trade (x-axis), and vice versa. There are also exceptions to this pattern, as we detail in the manuscript: this is the nature of optimal transport, which describes a spatially correlated system where finite supply is distributed among the destination countries.

5. *The authors should corroborate the results on the C figures with simple Spearman correlation coefficients to understand the underlying relationship between change in cost and change in trade.* We thank the reviewer for this interesting remark, and have added several additional plots to the SI substantiating our observations on the macro trends in the wheat trade following the war in Ukraine—see figs. S5, S7, and S8. Figure S5 shows the change in cost ΔC and relative change in trade volume $\Delta T/T$ for the period 2021–2022 as a function of GDP/capita (in real terms), as well as the Spearman correlation coefficients. We find a correlation coefficient of -0.23 for ΔC , indicating that poorer countries saw higher increases in cost than wealthier ones. We also show changes in cost as a function of relative changes in trade in figure S7, disaggregated by region. We see again that Sub-Saharan Africa bore a disproportionately high increase in trade costs, despite seeing similar changes in wheat imports as Europe, North America, and Oceania. See also the newly added discussion in the SI.

6. Gravity model (GM):

- (6.1) *Codes and data for the gravity model experiments are missing in the Github repository provided.* We thank the reviewer for noticing this; indeed we had omitted to push the results to repository, which we have now done. All plots can be reproduced using the `Evaluate.ipynb` Jupyter notebook. The Gravity model covariates and code are given in the `data/Gravity_model` folder.
- (6.2) *Eq. (20) does not make sense as it is currently formulated: the variables O_i , E_j , χ_j would be absorbed into the fixed effects. Since Table S1 reports coefficients (which would not be defined if the specification was in the current form), I suppose that the authors are really specifying one model for each good, i.e. $\log T_{i,jl}$ where l is the product.* We thank the reviewer for pointing out this inconsistency in the model specification. Yes, the gravity model is specified for each product l , separately, and we have now clarified our notation. In the original specification, O_i , E_j , χ_j were meant to indicate time-dependent variables; we have clarified the notation to include time and product indices.
- (6.3) *Why isn't the comparison with the GM also performed with goods fixed effects (either good specific γ_l or interacted with origin and destination). I would think the latter to be a fairer comparison due to the stringency of the NNs approach, cf. the discussion on multilateral resistance terms in the gravity literature.* We believe this point is a result of our confusing choice of notation, see Q1 above. In fact, we do estimate the GM separately for each product, and hence the estimated effects are effectively interacted by product by definition.
- (6.4) *A similar argument applies also to the time dimension. First, is not clear whether the comparison is done on a single year (cross-section) or for all years together (pooled sample). Then, if the NN is trained using all the data, then also in this case I believe a solid comparison would require the inclusion of time fixed effects.* Thank you for this relevant and important question. The comparison is done for the pooled sample. Our specification included origin, destination, and time fixed effects. We have now rewritten the GM section of the Supplementary Material for an accurate representation of the specification.
- (6.5) *A follow-up point from the previous one would be also a comparison in terms of the number of parameters of the two models. Indeed, one drawback of adding more fixed effects in the specification is that one should then estimate a lot of parameters. This would of course give a much better fit and fewer degrees of freedom. How would the GM and the NN compare in terms of this? This should be a further dimension to discuss for a fairer comparison between the two approaches.* For this, we now report the number of parameters of the neural inverse optimal transport model and the base gravity model. As an extension, we have included the estimation of the three-way gravity model with exporter-time, importer-time and pair (exporter-importer) destination fixed effects, removing corresponding regressors. The results show general consistency between optimal transport and gravity modelling approaches with high dimensionality, with the optimal transport approach fitting higher value trade flows more closely. One can argue that the gravity modelling fit will get even better with further time-dependent bilateral trade regressors. However, the optimal transport approach provides trade cost estimates in the absence of such regressors. We raise these points in the discussion, and thank the reviewer for their comment.

Minor Comments

1. *The B images in Case Study 1 would require a different scale.* We are very grateful to the reviewer for pointing this out—the large values on the scale were in fact due to a typing error in the f-string setting the colourbar labels. We have corrected the error, but also changed the scaling of the colourbar to now hopefully make it easier to parse.
2. *I believe the reference 22 is more subtle than what reported—aggregation matters when parameters vary across sectors (akin to the case where good specific fixed effects are included)* We agree that our statement was not precise and therefore rephrased it more carefully.

3. I would also suggest the authors to move one of the figures (perhaps Fig. 2) of the case studies in the appendix. I think there is some graphical redundancy and one has to do a lot of back and forth between text and images. We thank the reviewer for this suggestion, which we have followed.

A Soya dataset

B Barley dataset

(continued next page)

C Wine dataset

D Wheat dataset

(continued next page)

E Beef dataset

Figure 1: Splitting the FAO data into a train and test period (pink shaded area), and predicting trade shocks. The left two columns compare the FAO values (x-axis) with our estimates (y-axis), colour-coded by relative error. Also indicated are the median relative error (MRE) and the Pearson correlation on the entire dataset. The right side shows predicted trade values on a selection of corridors, with the test period (the years withheld as test data) indicated by the pink shaded period.

Reviewer 3

1. *To my understanding the authors take as given the demand and supply (parameter μ and ν) and solve for the optimal trade flows subject to these amounts. If that is right, then demand and supply are not endogenous and we are missing important equilibrium effects. This is particularly important for performing counterfactuals. We thank the reviewer for raising this important point, which helped us position our approach better. Our model treats the demand and supply as exogenous and estimates the bilateral trade costs that produces these trade flows. For counterfactual analysis, one can follow an approach similar to the conditional equilibrium analysis of Yotov et al. (2017), designing counterfactuals that lead to changes in trade costs. This can be done through regressing the trade costs on policy variables and then changing the levels of regressors according to the counterfactual scenario, e.g. the signing of a regional trade agreement. Extending the current approach to a full general equilibrium setting will require a complementary model of internal production and pricing, which is beyond the scope of this study. We note this as a promising research direction.*
2. *I am not sure full flexibility is desirable. The reason is the following: allowing for a fully flexible matrix of trade costs makes me worry about the validity of this specification in counterfactuals. In other words, it is less convincing that these trade costs will be invariant in counterfactual predictions, especially if the counterfactual is "far from our data" It is important to note that this model has not been designed for prediction. Building a predictive model of trade-flows would be uniquely challenging due to the sheer number of exogenous factors, feed-back delays and non-stationarity. Our model's objective is inference of an underlying cost structure which is compatible with the partial and noisy observations of trade-flows, which can provide unique insight into emerging trade barriers and corridors. One can also analyse the impact of changes in trade costs on trade flows, but a full general equilibrium analysis is beyond the scope of this study, as discussed above.*

3. *It was not clear to me how the authors performed the empirical analysis. It would be helpful if the authors explained this a little better. For instance, related to the above: it is reassuring that the proposed methodology can better predict what happened after the specific shocks studied; however, it seems to me the authors use data that include these shocks. It would be better to do this as an out of model fit type of exercise.* The empirical analysis was carried out by looking at changes in trade costs for each commodity and comparing it to the change in trade volumes. For instance, in the case of the Ukrainian war, we look at the change in cost and the associated (relative) change in wheat imports. The reviewer is right in thinking that the shocks are part of the training data—we use the entire set of FAO trade matrices, chiefly because data is very scarce and the quality of the trade matrices low (i.e., exporter- and importer-reported values do not agree, and we assume that a substantial number of unreported values are in fact non-zero, see fig. 4). For this reason, we were not optimistic that we would be able to extrapolate to unseen data, and especially predict the strong, out-of-sample shocks we analyse, by training on a small number of reported trade matrices. Nevertheless, we ran the numerical experiments as the reviewer suggested. To do so, we choose five datasets—beef, barley, wheat, soya, and wine—and masked four years containing some of the biggest shocks to the system (i.e., the periods of trade frictions: 2016–2020 in the case of soya, 2018–2022 for the others). We re-trained the neural networks and compared the performance on the test and training set. Overall, we observe between 1% and 21% drop in performance (measured as the Pearson correlation on the test data), and a significant increase in the median relative error (MRE) (see figs. 1). Some of the shocks are reproduced well, e.g. the drop in US-China soya trade, and the following spike in imports from Argentina; others, such as the drop in Australian beef exports to China, are not captured. Performance also varies considerably between the exporter- and importer-reported data. In order to conduct further analysis for improving out-of-sample predictions, one could look into training the model with many other commodities from UN Comtrade or BACI databases that include trade shocks of such high magnitude—but this is perhaps best left for future study. In this work, we are concerned with inference, and demonstrating that the inferred cost structure provides an informative and insightful perspective on the hidden drivers of trade.
4. *The regularization converts the standard linear LP optimal transport problem into a nonlinear one. The resulting maximum entropy approach and its economic meaning must be further explained considering the role of the hyperparameter ε , which needs further elaboration in connection with the case studies also.* In the economic literature entropic regularisation is often interpreted as a heterogeneity that is unobserved by analysts. We expect that trade flows are mainly determined by matches based on the underlying cost. Therefore the hyperparameter ε is chosen to be relatively small in all computational experiments—see also response to Reviewer 4 Q2.
5. *The duality in (17) and (18) refers to the LP version. The authors should explain the dual version of the nonlinear regularized version.* We originally intended to include the more ‘intuitive’ formulation for readers not familiar with the OT literature. The referee is however correct—the stated problem is the dual of the classic OT problem. We have corrected this and now state the correct dual problem, its connection to the classic OT problem as $\varepsilon \rightarrow 0$, and discuss its interpretation.
6. *Training is based on random batches of \mathbf{T} which produce ‘target’ cost values. The authors should elaborate further the behaviour outside the training set over a test set.* Please see the answer to question 3 above.

Reviewer 4

1. Can the authors comment on the well-posedness of this inverse problem? Is it generally true that there exists a unique \mathbf{C} such that the optimal transport plan is equal to an observed \mathbf{T} ?

The general solution of the entropy-regularised optimal transport problem is

$$\mathbf{T} = \text{diag}(\mathbf{u})e^{-\mathbf{C}/\varepsilon}\text{diag}(\mathbf{v}), \quad (1)$$

where u and v are scaling vectors. Taking (elementwise) logs and rewriting, we obtain

$$\mathbf{C} = -\varepsilon(\log \mathbf{T} - \log \mathbf{u} - \log \mathbf{v}^\top), \quad (2)$$

thus \mathbf{C} is only determined up to an additive decomposition into row and column vectors,

$$C_{ij} = -\varepsilon T_{ij} + \varepsilon \log u_i + \varepsilon \log v_j. \quad (3)$$

Any transformation of the kind

$$C_{ij} \mapsto C_{ij} + \alpha_i + \beta_j \quad (4)$$

(additive rank-2 transformation) leaves the transport plan \mathbf{T} invariant, since the transformation can be absorbed by the scaling vectors. Without further constraints, \mathbf{C} is therefore not uniquely inferable from observations of \mathbf{T} .

There are various additional assumptions one could make to ensure the problem is well-posed. We realised that our previous approach of requiring $C_{ij} \in [0, 1]$ and $\sum_j C_{ij} = 1 \forall i$ was in fact not sufficient to ensure uniqueness: a simple counterexample is the $N \times N$ matrix

$$\begin{pmatrix} 0, \frac{1}{N-1}, \dots, \frac{1}{N-1} \\ \vdots \\ 0, \frac{1}{N-1}, \dots, \frac{1}{N-1} \end{pmatrix}, \quad (5)$$

which has column sum 1 and whose associated transport plan is invariant under the transformation

$$\mathbf{C} \mapsto \mathbf{C} + (q, -\frac{q}{N-1}, \dots, -\frac{q}{N-1}), \quad 0 \leq q \leq 1. \quad (6)$$

Figure 2: Well-posedness of the inverse problem on synthetic data. We infer the cost matrix on noiseless, gapless, synthetic data, and require the diagonal of the transport plan to be mapped to 0, with each $C_{ij} \in [0, 1]$. This ensures shift invariance of the inferred \mathbf{C} and thus well-posedness of the inference problem.

Figure 3: Well-posedness of the inference problem on FAO data. Shown are results on the Soya bean dataset. We quantify uncertainty both by pushing the uncertainty on the training data (the transport plans) through the neural networks, and by training an ensemble of neural networks to measure the spread in possible cost matrices that minimise the loss function. Top row: we show the inferred transport plans, alongside the FAO data. Second row: each line represents the mean predicted cost matrix by a single neural network instance. We perform a sweep with 10 neural networks trained in parallel. As can be seen, the cost matrix is essentially uniquely inferred, though with some variation due to the optimiser getting trapped in local minima. Third row: the uncertainty on our new estimates covers the uncertainty both from the discrepancy between importer- and exporter-reported values, and the potential non-uniqueness of the optimal cost matrix. Since we are no longer constraining the column sums to be 1, the new cost estimates are overall higher than our previous ones (bottom row), though they are mostly qualitatively similar. See also fig. S4 in the appendix, and the newly added discussions on uncertainty quantification and well-posedness in the main manuscript and SI.

A natural way of making the problem well-posed is the following: we relax the column-sum constraint and instead require that the neural network

$$u : \mathcal{T} \mapsto \mathcal{C} \quad (7)$$

maps 0 to 1:

$$u(0) \stackrel{!}{=} 1. \quad (8)$$

This is reasonable: it simply says that the cost on an edge with zero flow should be maximal. If the maximum is

attained in *every* row *and* every column of \mathbf{C} , then the row- and column-shifts α_i and β_j must satisfy

$$1 = \max_j C_{ij} \stackrel{!}{=} \max_j (C_{ij} + \alpha_i) \leq 1 \forall i \Rightarrow \alpha_i = 0 \quad (9)$$

(and similarly β_j). Thus, the cost matrix is uniquely determined under these conditions (see fig. 2/fig. S1 in the SI).

The FAO transport plans satisfy this condition: they contain zeros in every row and column, since the vast majority of edges have zero flow; and even if a source country were to export to *all* destination countries, we could simply add that source as a destination and set the flow to 0, and vice versa. We have therefore modified the loss function used to learn the cost matrices to

$$J = \|\hat{\mathbf{T}} - \mathbf{T}\|_2^2 + \eta \sum_{(i,j) \in \mathcal{S}} (C_{ij} - 1)^2, \quad (10)$$

where $\mathcal{S} = \{(i, j) \mid T_{ij} = 0\}$. η is a balancing parameter that weights the regulariser in the loss term, and should be set such that the regulariser does not outweigh the term $\|\hat{\mathbf{T}} - \mathbf{T}\|_2$.

This shows that the inverse problem is unique when the forward optimal transport problem has a unique solution; however, with the FAOStat data, the relevant question is whether the loss function J has a *unique minimiser*. To account for this, we now train an *ensemble* of neural networks on each dataset, and compare the predictions. Overall, the various runs agree well with each other (see fig. 3, second row), indicating that our constraint allows us to extract a unique cost ‘signal’ from the transport plans by training a single neural network. The differences are in part due to the network not finding optimal cost matrices in every run; but there will also be uncertainty due to non-uniqueness of the optimal cost. By combining ensemble training with using the neural network as a pushforward measure, the uncertainty on the costs will now better reflect both the degree to which the inverse problem is ill-posed, as well as the uncertainty on the FAO data—see also the newly added fig. S4 in the SI. We have added a thorough discussion of these questions to the main article and the SI, and sincerely thank the reviewer for helping us close this important gap.

2. *How robust is the method with respect to missing data in \mathbf{T} ?* We have performed a number of additional experiments to analyse robustness with regard to missing values of \mathbf{T} (fig. 4.) We masked a random number of entries in the transport plan, and re-infer the cost matrix \mathbf{C} , given the marginals of the unmasked transport plan. The method is robust for small amounts of missing data, since the number of gaps in each row and column will be small, thus constraining how the missing ‘mass’ given by the marginals can be distributed among the missing entries. The error on the non-masked values of \mathbf{T} remains approximately constant (which is unsurprising), while the error on the entire cost matrix increases with the masking fraction. The error on \mathbf{C} increases linearly with the masking fraction, but remains significantly smaller on the unmasked edges (red).

For the FAOStat data, we previously considered all non-reported values as ‘missing’, giving us a masking fraction of around 80%. This is unreasonable: most values are not missing, but simply unreported zeros. We have thus re-estimated the number of missing values in the following way: if a value is reported by *any* side (importer or exporter) *either* for the quantity dataset *or* the value dataset, we consider that edge to be a non-zero edge; if no values are reported by *any* reporter, we set the flow to 0 and require the cost to be maximal on that edge. This way, we estimate the number of missing data points to be on average around 20% (see fig. 4).

3. *The parameter ε in equation (10) appears to be a hyper-parameter. How does ε affect the well-posedness of the inverse problem, and more importantly, how does the inferred \mathbf{C} depend on ε ? The authors may wish to demonstrate some robustness of their results with respect to the choice of ε , or discuss how this parameter should be chosen in a reasonable way. Similarly and perhaps less importantly, the number of iterations of the Sinkhorn algorithm in equation (19) (definition of $\hat{\mathbf{T}}$) is also a hyper-parameter. Please also discuss its effect and associated robustness. ε is a scaling parameter*

Figure 4: Top row: we infer the cost matrix on noiseless synthetic data with different fractions of the transport plan masked. Middle row: inferred transport plans. The ground truth is the same as in figure 2. Bottom row: As the fraction of masked values increases, the average L^1 error on T and C increases (blue). However, on the non-masked values, errors remain significantly lower, and the prediction on the training values of T is independent of the proportion of masked values (red). Shown are mean and median values, as well as the standard deviation, over all entries of the respective matrices. Right: the estimated fraction of missing values in each FAO dataset. This is estimated by comparing the number of entries reported by one reporter (exporter/importer) but not the other.

that determines how much small costs affect the transport plan (fig. 5). It can take any value in $[0, 1]$ (in principle, it can take any value in \mathbb{R}_+ , but for large ε the problem becomes ‘meaningless’ since all structure in C is blurred out). One would like ε to be as small as possible, since for $\varepsilon \rightarrow 0$ the inference procedure converges to classical OT. However, as $\varepsilon \rightarrow 0$ Sinkhorn’s algorithm becomes unstable, because as the entries of the initial guess $\exp(-C/\varepsilon)$ go to 0, the required scaling vectors need to grow exponentially to match the marginal constraints. This also causes

Figure 5: Effect of the entropy regulariser ε on the inference. Top row: the transport plan for the cost matrix given in 2 but different values of ε . Bottom row: accuracy on the inferred transport plan and cost matrix as a function of the regulariser ε . The cost matrix is the same as in fig. 2. Each line is an average over 5 different seeds.

the convergence rate of Sinkhorn’s algorithm to slow significantly as $\varepsilon \rightarrow 0$, increasing the computational cost. We therefore choose a small value of $\varepsilon \approx 0.1$ that balances numerical stability and computational cost. The inference accuracy is independent of the choice of ε —see fig. 5. In principle, ε could be inferred from data so long as both the cost and ε are constrained to lie in the unit interval; however it is not immediately clear to us how this affects the uniqueness of the minimizers on noisy data, and so we leave this question for future research.

Regarding the number of Sinkhorn iterations, these are in fact not fixed, but rather a tolerance criterion is used to determine when the algorithm should terminate: let \hat{u}^k and \hat{v}^k be the scaling vectors in iteration k of the Sinkhorn algorithm; then we terminate the algorithm if $|u_i^k - u_i^{k+1}| < \eta \forall i$ and $|v_j^k - v_j^{k+1}| < \eta \forall j$, where $\eta = 10^{-9}$. This effectively means we terminate the algorithm once it has converged. We thank the reviewer for this observation and have amended the text accordingly.

4. *The results here are not a demonstration of whether the inferred \mathbf{C} is more accurate. Do the authors have any way of demonstrating that the inferred \mathbf{C} is correct, perhaps on a synthetic example?* See response to point 1 and figures 2.
5. *One would expect that with a greater number of well chosen covariates in the regression below equation (1), one may get better reconstruction for the gravity model. Have the authors ruled out that this is due to the small number of (or badly chosen) covariates that cause the reconstruction performance of the gravity model to be sub-optimal?* We thank the reviewer for raising this point. The baseline gravity model’s inferior performance is almost certainly due to a sub-optimal choice of covariates. The large spread of the gravity model’s accuracy between commodities (fig. 6C in the manuscript) would generally indicate that the chosen covariates are a good choice for some commodities

and a bad one for others; the OT approach, meanwhile, performs consistently well. Hence, our approach does not rely on a prudent choice of covariates and functional form for accuracy. Since the cost function is not being parametrised, one does not need to find the ‘right’ covariates in order to model the system well. This comes at the cost of direct interpretability. However, the learned cost structure can thereafter be modelled using a selection of covariates, should one wish to interpret it more directly. Additionally, we have now added a three-way gravity model, which has high-degree fixed effects (see also responses to Reviewer 1) and closes the gap between the degrees of freedom of the gravity model and OT. Our approach provides an alternative approach by estimating trade costs purely from trade flow data.

6. *Does the current approach train a different cost \mathbf{C} for each traded commodity? If so, does it make sense to account for correlations on \mathbf{C} for different commodities instead?* Indeed, the current approach trains a different cost for each commodity independently. We agree that introducing learned correlations between commodities would be beneficial and potentially illuminating; but while this is certainly possible, introducing it would require a substantial extension of the current methodology, and so we leave this for future work.
7. *Page 3: “both prediction accuracy and uncertainty”, do you mean “uncertainty quantification”? Where is it demonstrated that uncertainty quantification performance of the OT approach is also superior to that of the gravity model?* Indeed, we have not compared our method’s uncertainty quantification to that of another method, and have thus amended this part. Our method provides uncertainties on \mathbf{C} that reproduce the errors on $\hat{\mathbf{T}}$ given by the FAO data—but that is all. We thank the reviewer for pointing this out.

Minor issues

1. *After Equation 5: you may wish to cross reference the definition of \mathcal{H} in methods, since it is the negative entropy, not the entropy—this may confuse some readers.* Corrected.
2. *Equation 8: the notation here is rather confusing as u is a function mapping transport maps to costs as defined in equation 7. If the LHS is a probability density function over costs, then the RHS should be something like*

$$\rho(u^{-1}(C)) \left| \det \frac{\partial u^{-1}}{\partial T}(C) \right|? \quad (11)$$

We thank the reviewer for flagging this: our statement is indeed a little loosely formulated. What we are describing is a pushforward measure,

$$\rho(\mathbf{C}) = (u_{\#}\rho_T)(\mathbf{T}), \quad (12)$$

where ρ_T is the measure on \mathbf{T} . This is well-defined even if the neural network is not invertible (which in general will not be). What the reviewer is describing is a change of variables transformation, which can in principle be defined if u_{θ} is smooth (which it is), differentiable (which it is) and locally injective, but this is not required to define the pushforward. We have amended the text to avoid this reading. We obtain $\rho(\mathbf{C})$ as before through sampling from ρ_T , though in addition we use ensemble training to account for the potential ill-posedness of the inverse problem.

3. *In the analysis on case studies the trading cost changes are reported in “units”. Can the authors explain what units are these, and how one should interpret its magnitude?* The cost matrix is unitless, and can only ever be understood as a measure of relative cost. Hence, any individual value C_{ij} only confers meaning as a relative measure of cost, ‘relative’ here meaning with regard to the rest of the matrices, or as a time series $C_{ij}(t)$. We have clarified this point in the manuscript.

Modelling Global Trade with Optimal Transport

Response to reviewers

We would like to thank the reviewers for their feedback on our manuscript. We have revised the manuscript and code according to their suggestions; please see a point-by-point response attached.

We look forward to hearing from you,

Yours sincerely,

Thomas Gaskin, Guven Demirel, Marie-Therese Wolfram, Andrew Duncan
9 December 2025

Reviewer 4

1. *The authors clarified that the Sinkhorn algorithm is in fact not run for a fixed number of iterations but run till a tolerance is reached. In this case, the number of Sinkhorn iterations would also be a (discrete) function of C . This would pose some issues when you back-propagate to compute the derivative of \hat{T} with respect to C , as the function is not differentiable at points at which the number of iterations jump. This is probably a minor in terms of numerics, but it would be good to be careful at least in the explanation of your approach.* While the number of Sinkhorn iterations depends discretely on the input, in our implementation the gradient is computed by differentiating through the executed iterations only. Each Sinkhorn step is a composition of smooth operations (exponentials, normalisations), so the map is differentiable *conditional on a fixed iteration count*. The dependence of the stopping time on C indeed introduces potential non-differentiabilities at iteration-change boundaries; however, in practice these events are rare and did not cause instability in our experiments. Additionally, one can avoid this issue entirely by running Sinkhorn for a fixed, sufficiently large number of iterations, which yields a fully differentiable computation graph and is a common practice when strict differentiability is required (possibly at the expense of computational speed). We have added a note to this effect.
2. *I suggest hosting the data folder at some hosting site (e.g. huggingface, Harvard dataverse, etc) so that one is not forced to download the data when cloning the repository (it's a few GB). The requirements is not complete, e.g. `geopandas` is missing, `netcdf4` is missing (both throw errors in the notebook - and I cannot seem to load the dataset in cell 4, may have something to do with git LFS). I suggest the author double check the repository using a clean environment and see if all the steps can be reproduced successfully.* We thank the reviewer for going through the code so carefully: we have updated the requirements.txt file with `geopandas`. The `netcdf4` backend was not loading since our git LFS quota had been used up and the data could thus not be downloaded. We have therefore moved all the data to a huggingface repository and linked it in the document. The README has been adjusted accordingly. We have checked the code on a clean machine, and have also tested it against five Python installations: (Python 3.9–3.13).